# Leveraging Quality Improvement and Shared Learning to Improve Infant Well-Child Visit Rates in Texas

**DOI:** 10.3390/healthcare12191965

**Published:** 2024-10-02

**Authors:** Emily Stauffer Rocha, Susana Beatriz Peñate, Ryan D. Van Ramshorst

**Affiliations:** 1Office of the Medical Director, Medicaid and CHIP Services, Texas Health and Human Services Commission, Austin, TX 78751, USA; ryan.vanramshorst@hhs.texas.gov; 2Office of Policy, Medicaid and CHIP Services, Texas Health and Human Services Commission, Austin, TX 78751, USA

**Keywords:** medicaid, infant well-child, well-child visit rate, infant health, quality improvement, learning collaborative, PDSA cycle

## Abstract

Texas Medicaid improved infant well-child visit rates by participating in a national learning collaborative. The two-year program encouraged creativity and innovation in care for Medicaid recipients through partnerships with managed care organizations (MCOs). The MCO projects discovered valuable practices in member outreach and were disseminated in shared learning experiences. At the completion of the learning collaborative, Texas Medicaid surveyed the MCO participants to assess the impact of their projects on Medicaid beneficiaries in Texas as well as the quality improvement project format. Collectively, the MCOs raised the infant well-child visit rate year-over-year. All of the partner MCOs stated they plan to continue focused work on improving infant well-child visit rates after the learning collaborative.

## 1. Introduction

Children who attend well-child visits are more likely to be up-to-date on immunizations, have developmental concerns recognized early, and are less likely to visit the emergency department [1]. Unfortunately, the national rates of preventive medical visits have decreased notably, a net 5% decrease between 2016 and 2020, according to an article from the Journal of the American Medical Association (JAMA) [2]. Moreover, the percentage of children nationwide receiving six or more well-child visits in the first 15 months of life was only 64 percent in federal fiscal year 2019 based on data from the Centers for Medicare and Medicaid Services (CMS) [1]. The recent COVID-19 pandemic only exacerbated the issue of children’s healthcare use declining after the pandemic’s onset [2].

Differences in the rates of well-child visits and preventive care exist based on a child’s insurance coverage. Medicaid was signed into law in 1965 and allows all states, districts, and territories to provide health coverage for low-income people [3]. The Children’s Health Insurance Program (CHIP) was signed into law in 1997 and provides states with federally matched funds to provide health care coverage to children who cannot afford private health insurance and who do not qualify for Medicaid coverage [3]. Children covered by public insurance, such as Medicaid and CHIP, have lower rates of attendance at well-child visits than children covered by private insurance [4]. Additionally, during the first year of life, healthcare utilization among Medicaid-insured infants (not preventive care related) is greater than that among commercially insured infants [5]. State Medicaid programs have long covered well-child care and preventive services, but a renewed sense of urgency to improve outcomes arose with the onset of the pandemic.

CMS and Mathematica, a quality improvement organization, launched the Infant Well-Child Visit Learning Collaborative to support state Medicaid teams in improving the rates of well-child visits. The learning collaborative launched in 2021 and ran for two years, ending in October 2023. The CMS and Mathematica teams provided technical assistance to state Medicaid and CHIP agencies to create projects focused on increasing the well-child visit rate for beneficiaries 0–15 months of age [1]. Suggested strategies to improve well-child visit rates included using data to identify disparities and increase participation in well-child visits, engaging families, healthcare providers, and partner agencies, using payment and incentives, and leveraging evidence-based models of care [1].

Texas Medicaid and CHIP Services (MCS) was one of six state entities selected to participate in the learning collaborative. Over the two-year project span, MCS partnered with ten managed care organizations (MCOs) to implement quality improvement projects aimed at improving the organizational rates of infant well-child visits. The projects utilized the plan-do-study-act (PDSA) cycle format to test small-scale interventions over short periods of time. With every PDSA cycle, MCO teams collected data and analyzed outcomes prior to performing subsequent cycles. With technical assistance from the MCS team, MCOs modified interventions or populations with each cycle. PDSA cycles were able to be altered based on outcomes to achieve efficiency and value in their processes.

Throughout the project, the MCS team held learning sessions where the MCO partners had the opportunity to share their progress along with lessons learned and key takeaways. MCOs were encouraged to ‘share seamlessly and steal shamelessly’ to improve their own projects. At the end of the two years, the MCOs shared their plans for expanding and implementing their projects to ensure continued improvement in infant well-child visit attendance.

The MCS team gathered infant well-child rate data throughout the project and at the conclusion. Each partner MCO was responsible for collecting infant well-child rates during the project. The MCS team combined the rates of the partner MCOs into a project-specific visit rate, which was monitored on a monthly basis. A survey of the partner MCO teams at the conclusion of the learning collaborative was conducted to collect information on outcomes from each organization’s quality improvement project, demographics of their project population, and feedback on the learning collaborative. 

## 2. Materials and Methods

The survey conducted at the end of the learning collaborative included questions aimed at gathering demographic data about the MCOs, project design, outcome data, and input on the feasibility and perceived impact of the project. The survey was created using Microsoft Forms, and a link was emailed to the partner MCO teams. Teams were able to submit their responses anonymously and had approximately one month to respond. The answers were reviewed by the MCS team and analyzed for key themes. The information from the survey will be used to aid MCS in supporting MCOs during future learning collaboratives.

## 3. Results

The infant well-child visit rate was the primary outcome metric of the project. The well-child visit rate metric is standardized by the National Committee for Quality Assurance (NCQA), specifically the Healthcare Effectiveness Data and Information Set (HEDIS). The well-child visit rate has two parts, one of which focuses on the first thirty months of life [5]. This metrics “assesses children who turned 15 months old during the measurement year and had at least six well-child visits with a primary care physician during their first 15 months of life” [6]. The metric is calculated to include the prior eight quarters of data, making changes in outcomes visible only when assessing year-over-year data and like months.

Each MCO was responsible for submitting monthly visit rate data to the MCS team. These monthly visit rates were combined into a project-specific data point (Figure 1). Months with input from the majority of MCOs were compared to assess the success of the project. Months with incomplete data from MCOs, specifically January, February, and March, were not included. Due to year-end data calculations and the cumulative nature of the metric, starting with the complete month (April) does not impact the assessment of outcomes. The table below shows average infant well-child visit rates for the MCOs that participated in the project. The baseline year was set to 2019 to mitigate any effects of the COVID-19 pandemic. The study years were 2022 and 2023. The table shows an improvement in infant well-child visit rates compared to baseline as well as from the first study year to the second study year. While increases in infant well-child visit rates occur as time progresses in the calendar year, the true improvement is seen in the month-to-month comparison. The partner MCOs achieved higher visit rates in like months compared to the baseline year. Additionally, visit rates climbed more quickly in the intervention years compared to the baseline year, suggesting more children attended well-child visits on time. The end of the year rate for the baseline year and the second study year were similar, suggesting the project allowed partner MCOs to overcome COVID-era impacts.

The survey questions captured information about the MCOs and their member populations, project design and outcomes, and perspectives on the PDSA cycle project methodology. All the partner MCOs offered at least one of the child-serving Medicaid managed care programs in Texas (STAR [7] and STAR Kids [8]), and some offered CHIP. Half of the organizations described themselves as large, meaning they provided services in multiple Medicaid service delivery areas (SDAs). The other half described themselves as small, meaning they provided services in only one or two service delivery areas. Between the partner MCOs, all service delivery areas in Texas were covered, though projects may have only focused on one or a few SDAs.

Most of the MCO partners stated their population of 0–15 month-olds made up less than 25 percent of their members. One MCO had 26–50 percent of their membership in the 0–15 month-old range, and one MCO had over 75 percent of their membership in the 0–15 month-old age range. Based on responses from MCO estimates, the outreach conducted to members by all MCO projects throughout all PDSA cycles impacted approximately 15,000 children. 

The population of focus for each project varied in the type of geographical areas where members lived. MCOs described the areas where members lived as urban (8), suburban (4), and rural (6). Projects focused on different geographical areas during different cycles, or MCOs decided to change the area of focus at some point during the project based on findings. Populations of focus for seven projects included all races and ethnicities. Two projects focused on the Black, non-Hispanic population for at least one PDSA cycle. Two projects focused on the Hispanic/Latino population for at least one PDSA cycle. Two projects focused on children with medical complexity for at least one cycle. 

Most MCOs conducted more than ten PDSA cycles. One MCO was conducted between six and ten PDSA cycles. Three MCOs conducted one to five PDSA cycles. PDSA cycle length varies by MCO and sometimes from cycle to cycle. Half of the MCOs conducted one PDSA cycle per month. Three MCOs conducted one PDSA over multiple months. Two MCOs conducted each PDSA in a two-week period.

All MCOs used phone calls as a method of outreach to members. Five MCOs also used traditional mail, three used email, two used text messaging, and two used another form of communication (Figure 2). Most MCOs conducted outreach utilizing MCO staff. Four MCOs partnered with a provider office or third-party vendor to conduct member outreach.

In addition to monitoring infant well-child visit rates, MCOs were tasked with developing and monitoring metrics that more closely identified changes in the population. The post-learning collaborative survey asked MCOs to describe their outcome metrics, process metrics, and balancing metrics that were tracked and used to amend PDSA cycles as needed (Table 1). The outcome metric for all MCOs was the same, as was the well-child visit rate, but it may have focused on specific subpopulations. The process metrics employed by MCOs varied based on project design. Balancing metrics, which include changes not initially anticipated to be impacted by the projects, varied by MCO and offer insight into future project planning. The below table describes some of the metrics monitored by the MCO teams. 

The outcome metrics were similar across all MCOs, with one MCO choosing to add an immunization metric. The outcome metrics allowed MCO teams to see movement in metrics due to interventions that the standardized well-visit metric could not. Most MCOs focused their process metrics on items that improved outreach and created consistency amongst staff. Some MCOs used the same process metrics but changed items, such as the timing of phone calls to members. Balancing measures were helpful to individual MCOs in that they offered opportunities for collaborating or improving clinic efficiency.

MCOs were asked in the survey to describe the outcomes of their chosen communication method(s), specifically lessons learned that could not be easily ascertained from data analysis. The table below includes lessons learned organized by communication method (Table 2).

The most popular outreach method used by MCOs was phone calls due to their efficiency and low cost. Phone calls allowed many MCOs to accomplish the task of scheduling children for well-visits as well as gathering additional information. Many members discussed barriers to care or additional needs that MCOs could help resolve. Text messaging was of interest to all MCOs, but only one MCO was able to implement a text messaging system during the project. All MCOs showed interest in text messaging with members and planned to implement a program in the future.

The post-learning collaborative survey asked MCOs to reflect on their focus population to determine the overall outcome of their projects based on what improvements were observed. The table below describes the improvements each MCO reported (Table 3).

Each MCO was asked to share improvements created by their project that may not have been reflected in their data. MCOs stated they felt interventions performed by the MCO staff were successful in increasing well-child visit rates and easily adapted from project cycle to project cycle. All MCOs saw increases in their well-child visit rates from the beginning of the project to the completion.

In addition to the overall outcomes of each project, the state team wanted to know what key lessons the MCO organizations learned about their focus populations. Specifically, the survey questions asked what was found to be helpful in getting kids to their well-child visits. These lessons learned are ideas that can be shared with other organizations interested in increasing their well-child visit rates. The table below describes key concepts for increasing well-child visit rates (Table 4).

This survey question was aimed at answering questions that may help other MCOs implement similar projects. Each MCO provided input and practical advice, such as asking about and addressing barriers to care or educating about value-added services. Most MCOs saw success when multiple inventions were combined.

Lastly, the survey included questions about any barriers the MCOs discovered, as it was important to learn if there were hindrances to well-child visit attendance. The barriers or challenges are described in the table below (Table 5). 

The most common barrier for MCOs was being able to reach their members. Despite this, phone call outreach was still the most efficient method of outreach. Many MCOs implemented processes or improved information collection based on their experiences in the project. 

All of the MCOs responded in the post-learning collaborative survey they would be continuing efforts to improve infant well-child visit rates after the conclusion of the affinity group. Most MCOs plan to scale up their projects. Input on what some organizations plan to do is described in the table below. (Table 6).

All MCO teams stated they would continue projects aimed at improving well-child visits at their organization. Some projects received funding at the organizational level to further train staff or create dedicated departments. Many MCOs expressed interest in creating text messaging campaigns or processes. Some MCOs wanted to expand efforts or try interventions in different aged populations, such as toddlers, school-aged children, or teenagers.

## 4. Discussion

The projects found multiple practical lessons that can be widely used by organizations to increase well-child visit rates. The first lesson was how communication with members was key to increasing well-child visit rates. There was no clear advantage to any one method of communication. Multiple methods of communication used in conjunction were found to be valuable and successful in the projects. The most important concept was to stay connected to members using the method that worked best for them. Meaningful communication was also achieved or bolstered through partnerships with provider offices. 

Next, there were pros and cons to partnering with a provider office or clinic. Through the use of electronic health records and texting services, partnering with provider offices provided MCOs with additional options for member outreach. Provider offices have different regulations surrounding text messaging capabilities, in contrast to MCOs, which are required to follow federal and state regulations. Challenges to partnering with a provider office included staffing and project modifications. Staff turnover in provider offices caused issues such as delays or pauses in implementing the project intervention. MCOs that partnered with provider offices were slower in modifying interventions from cycle to cycle.

The next lesson was that there is a higher likelihood that children will attend at least six well-child visits if they start attending well-child visits within the first few weeks or months of life. Organizations were able to bring members into compliance with the well-child visit rate metric when interventions focused on the early weeks and months of life. A few organizations focused on populations who were close to aging out of the metric and also had success with bringing those members into compliance. It was observed that members who attended visits early were able to attend more than the six necessary visits and that parents and guardians saw value in continuing to attend visits. 

Lastly, the PDSA cycle format allowed for rapid testing of a variety of interventions in a variety of populations. Organizations that were flexible and able to quickly modify their projects based on findings had the most success with the PDSA cycle format. Many MCOs created sustainable processes that could be scaled at the conclusion of the learning collaborative. Also, MCS fostered a shared learning environment by hosting biannual learning sessions. The learning sessions gave partner MCOs the platform to discuss their projects, lessons learned, and best practices for increasing well-child visit attendance. Organizations took away new interventions and avoided pitfalls experienced by their peers. 

## 5. Conclusions

The two most important lessons learned by the partner MCOs included providing meaningful communication to members and reaching out to members early in life. The projects that achieved the most improvement in infant well-child rates implemented interventions that provided early communication to members. The MCOs and MCS hypothesized that parents or legal guardians who were engaged early in an infant’s life found value in continuing to attend well-child visits or developing trusting relationships with providers. Support provided in the form of phone call outreach or text reminders from either an MCO or a provider office was successful and may have contributed to the increasing well-child attendance.

The PDSA cycle format of rapid cycle intervention testing was an effective method to quickly improve quality measures. The processes tested utilized a fraction of resources and were implemented on a small scale, which optimized insight into needed preparations for larger-scale implementation. Permanent processes were implemented with confidence due to more realistic expected outcomes in long-term improvement in quality measures.

## Figures and Tables

**Figure 1 healthcare-12-01965-f001:**
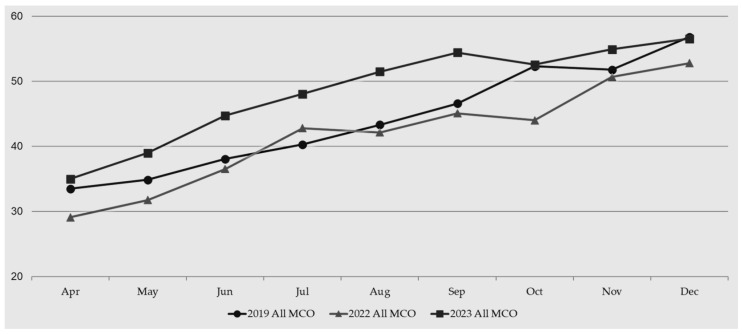
Average infant well-child visit rate for all partner-managed care organizations.

**Figure 2 healthcare-12-01965-f002:**
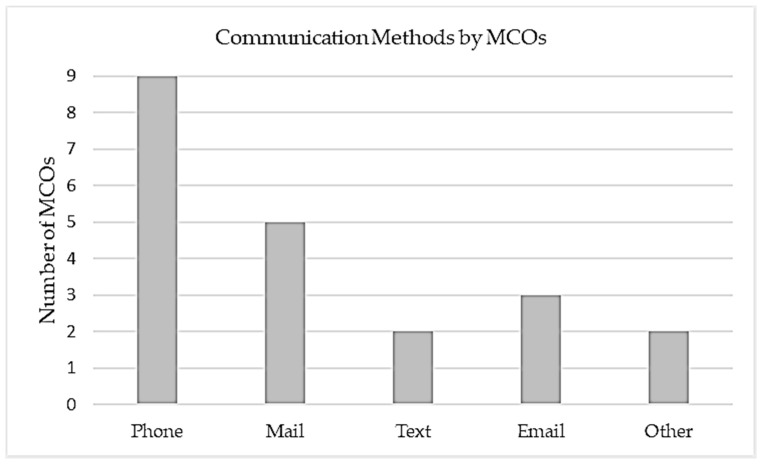
Methods of communication used in managed care organization projects.

**Table 1 healthcare-12-01965-t001:** Metrics monitored in multiple projects.

Outcome Metrics	Process Metrics	Balancing Metrics
Average number of visits in 15-month time period	Number of successful telephone outreach attempts	Improvement in office staff morale
Percentage of visits from 0 to 6+ years	Number of members who declined our assistance in scheduling appointments	Perceived increase in appointment length by parent or guardian
Monthly average and count of visits	Number of newly scheduled appointments that resulted from outreach	Perceived increase in waiting room time by office staff
Whether hospitalization delayed immunization.	Total members contacted	
	Members who completed their 4th through 6th visits	
	Members who had a visit scheduled	
	Number of days between discharge and immunization administration	
	Days between discharge and successful outreach	
	Number of members emailed and number of emails opened	
	Number of emails opened and click rate of embedded links	
	Number of educational materials provided	

**Table 2 healthcare-12-01965-t002:** Lessons learned by type of communication outreach.

Phone Call	Mailed Letter	Text Message
Phone calls to reachable members were effective.	Postal mail to unreachable members had a good response rate.	Text messaging was more successful [than any other outreach method].
Phone call outreach was most successful to members; multiple attempts were made when unable to reach member changing time of day and day of week.	Telephonic contacts with members were more successful than mailing out brochures.	
A phone call from MCO staff was most successful 24–48 h before scheduled appointment.		
Phone call outreach was the most successful for the STAR Kids population.		
More qualitative feedback was obtained [from phone call outreach].		
The phone calls were the most successful when we targeted members that were due for their exam. The phone calls were conducted on the population that did not have email addresses.		
Phone calls in urban areas had more successful outcomes.		

**Table 3 healthcare-12-01965-t003:** Observed improvements in focus populations ^1^.

MCO	Improvement
1	In 2022, we saw a 2% increase in 6+ visits in the target population. In 2023, we have seen approximately 1% increase YTD.
2	Our MCO saw an overall improvement with gap closures, we observed provider engagement and participation in contacting members who needed to complete open care gaps. The overall infant well-child rates are increasing month over month.
3	The rate of well-child visits increased more in selected members for each intervention with MCO [staff] doing the intervention versus the provider/provider offices doing interventions.
4	In Year 2, 6 of the 25 members are still eligible for the measure; 9 of them (36%) are now compliant.
5	STAR Kids showed an improvement in visit rates over baseline.
6	We found that making case management referrals while the member was still in-house shortened the days between discharge and successful outreach, as well as made improvements on the days between discharge and gap closure.
7	An increase in the rate of well-child visits was also reflected on W30, rate 2 (15–30 month-olds).
8	Based on our overall compliance, we saw a greater increase in compliance when we performed targeted outreach to members who were due for an exam vs. the generic emails that provided education on the well-child exams.
9	The post-outreach visits increased after successful phone calls in rural areas.

^1^ Actual results from partner-managed care organizations at the completion of the project.

**Table 4 healthcare-12-01965-t004:** Key concepts for increasing well-child visit rates ^2^.

MCO	Improvement
1	Early intervention is key. It is very difficult for members to get caught up when those initial few appointments are missed (especially for long-term NICU infants). Phone calls and letters are effective. Appropriately timed reminders are important.
2	We learned that parents will take children if they have time and are motivated by incentives. The parents are busy, they only take kiddos if they are sick, not necessarily for primary care.
3	We learned reminding members of our MCO value-added services specific to that well visit was helpful in getting the members in for the appointment, having good relationships with collaborating providers was beneficial in getting buy-in for them to participate in the interventions.
4	Providers were receptive to calling members. Caregivers are receptive to a call back to confirm if the appointment was kept.
5	In the STAR Kids population, focusing on educating caregivers on the importance of the relationship with their PCPs (primary care provider) was most impactful in getting kids to the [well-child visits]. Many of these members have complex care needs so caregivers prioritize specialty care over preventative visits. Surprisingly, transportation was not a significant barrier.
6	There are various reasons why members are not vaccinated. There were some whose providers did not have vaccines available, which our team were able to assist and make impact. There were others whose caregivers elected not to vaccinate as a personal choice. Education was provided to all regardless of choice with some resulting in caregivers deciding to delay schedule versus complete refusal. The most impactful lesson in getting kids to well-child visits were inquiring about the members’ PCP, and locating PCPs that the caregivers had autonomy to choose from a list provided.
7	Transportation and childcare was an issue for our members.
8	We learned that doing targeted outreach is more beneficial. Reaching out timely to members when they are due for a well child exam yielded better outcomes. Also members would benefit from sooner outreach to provide education and remind them of well child exams and the importance.
9	The number of post-outreach newborn well visits increased. The combined effort with the provider group in establishing reminder calls resulted in enhanced completion of well visits.

^2^ Actual input on key concepts from partner-managed care organizations at the completion of the project.

**Table 5 healthcare-12-01965-t005:** Barriers or challenges to well-child visit attendance ^3^.

MCO	Improvement
1	COVID-19 impacted staffing at provider offices and appointment availability. The “tripledemic” in the 3rd and 4th quarter of 2022 caused providers to delay well-visit scheduling to improve sick visit appointments and reduce the impact on emergency rooms.
2	Time, transportation, appointment availability, member education, lack of provider initiative, member contact (not answering, incorrect phone)
3	Barriers include member phone numbers on file for MCO and provider offices being incorrect or disconnected, and getting the list of selected members from each collaborating practice on time to be included with PDSA cycles.
4	Provider office limited to walk-in appointments. Lack of provider office staff. Lack of provider office processes (i.e., office did not do appointment reminders). Lack of availability with provider office. Inability to effectively track the attendance of our members in community-based events. Unable to reach member.
5	In the STAR Kids population, we found barriers related to NICU stays, impacting ability to complete six visits prior to 15 months of age; some members remained in hospital (i.e.,: NICU) for the entire W30, Rate 1 timeframe. STAR Kids members also have other health insurance which impacts our ability to monitor, track and/or account for well-child visits not billed to [the health plan]. For the STAR population, the inability to reach members/caregivers and the possibility that some members have changed providers and are no longer patients at [the provider office] were barriers we found during the initial outreach. Partnering with a large health system limited our access to individualized outreach results and also required flexibility with outreach priorities and scheduling.
6	Difficulty reaching members and their caregivers. Clinical [information] we received was not always accurate.
7	Members who did not answer the phone or where unreachable.
8	When we initially started this project, we wanted to text our members and were unable to, other than that there were no additional barriers.
9	Less successful phone calls in rural areas.

^3^ Actual input on barriers and challenges from partner managed care organizations at the completion of the project.

**Table 6 healthcare-12-01965-t006:** Post affinity group organizational plans.

Organizational/Existing Process Changes	New Processes/Services	Adding Member Populations
Create a wellness and prevention team	Implement text reminders and auto-dialer capabilities	Expand intervention to CHIP, STAR, STAR Health, and STAR Kids populations
Increase early application for SSI at 30 days NICU stay	Partner with marketing, quality, and provider relations departments	Expand to all members younger than 15 months
Identify all members with missed appointments and conduct outreach	Send member lists to providers	
Partner with community-based resources	Send text message before phone call to increase success rate of phone outreach	
Continue to collaborate with providers and provider office staff		

## Data Availability

The data analyzed in this study is presented and contained within this article. No additional data is available for sharing.

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
