# Peer review of "Leveraging Quality Improvement and Shared Learning to Improve Infant Well-Child Visit Rates in Texas"

_healthcare, 2024, doi:10.3390/healthcare12191965_

Round 1

Reviewer 1 Report

Comments and Suggestions for Authors

Many thanks to the authors for sending the manuscript. It is certainly a very good experience, and it gathers valuable information to improve the programs and the quality of care for children.
Given the information included in the article, the "Original article" format is probably not the most convenient, since a lot of information is missing:
- Abstract: it should have a balance between introduction, material and methods, results, and conclusion. Currently, there is a lack of clear information on methods and results
- In the material and methods section: a lot of information is missing. It is important to describe several very important aspects. For example, what did the program include? (what topics were addressed, how many sessions, in what format, to whom?) How was the target population selected?
On the other hand, what aspects did the evaluation include, and how were the participants selected?
- Were any ethical aspects considered?
- In the results, there is information that could be taken to methods, where they mention the differences that exist in the MCO during the implementation

Review the quality of figures 1 and 2.
In figure 1, the figure is not clearly understood. Most of the centers start in April with an average of approximately 35 visits per center/child? and in December they reach approximately 55? Is this difference significant? What could be the reason for these differences? It is not discussed
The same for figure 2. Could it be analyzed by center or characteristics of the program, to see a pattern?

The analysis of the “lessons learned” seems to have addressed qualitative information (focus group?), it is important to be clear about how the information was obtained and how it was analyzed

In the discussion, some aspects are mentioned that are a bit risky to mention, such as: (line 172) "There was no clear advantage to any one method of communication", since there are limitations in the analysis to make this statement

Author Response

Comment 1: Many thanks to the authors for sending the manuscript. It is certainly a very good experience, and it gathers valuable information to improve the programs and the quality of care for children.
Given the information included in the article, the "Original article" format is probably not the most convenient, since a lot of information is missing:
- Abstract: it should have a balance between introduction, material and methods, results, and conclusion. Currently, there is a lack of clear information on methods and results
- In the material and methods section: a lot of information is missing. It is important to describe several very important aspects. For example, what did the program include? (what topics were addressed, how many sessions, in what format, to whom?) How was the target population selected?
On the other hand, what aspects did the evaluation include, and how were the participants selected?
- Were any ethical aspects considered?
- In the results, there is information that could be taken to methods, where they mention the differences that exist in the MCO during the implementation

Response: Additional descriptive information was added to the sections mentioned in the comment to add clarity for readers.

Comment 2: Review the quality of figures 1 and 2.
In figure 1, the figure is not clearly understood. Most of the centers start in April with an average of approximately 35 visits per center/child? and in December they reach approximately 55? Is this difference significant? What could be the reason for these differences? It is not discussed
The same for figure 2. Could it be analyzed by center or characteristics of the program, to see a pattern?

Response: Figures 1 and 2 were adjusted for clarity. Additional clarifying information was added to the results section.

Comment 3: The analysis of the “lessons learned” seems to have addressed qualitative information (focus group?), it is important to be clear about how the information was obtained and how it was analyzed

Response: This is correct, the information was qualitative and is meant to give insight into other organizations wishing to duplicate success.

Comment 4: In the discussion, some aspects are mentioned that are a bit risky to mention, such as: (line 172) "There was no clear advantage to any one method of communication", since there are limitations in the analysis to make this statement

Response: A clarifying statement was added to the discussion section.

Reviewer 2 Report

Comments and Suggestions for Authors

The subject matter of this paper is interesting.  There are a number of problems with current content and presentation.

1) Arguably the conclusion that well-child visit rates rose is not actually supported by Figure 1.  Well-child visit rates in 2022 actually start and end lower than baseline (2019). 2023 starts off lower than 2022 ended and ends in the same place that baseline ended. 

Also, a few related issues:

             a) I understand why 2019 was used as baseline but perhaps it doesn't make sense.  After all, as you note, COVID had a big impact.

             b) Exactly how is well-child visit rate calculated?  That's not actually explained in the methods.

             c) Why does the rate always start off lower in April and rise towards December?  And why are January-March not shown?

              d) Was there variation in this outcome among the 10 MCO sites that may be useful to learn from?

2) Tables on what was learned through PDSA cycles have the greatest potential to offer insight, but currently there is no summary statement of what they say in the results.  There is some in the discussion, but summary should be moved to results and elaborated on. 

3) Small thing, but would be worthwhile to explain "balancing metrics" as some readers will not be familiar with that. 

Comments on the Quality of English Language

N/A

Author Response

Comment 1: 1) Arguably the conclusion that well-child visit rates rose is not actually supported by Figure 1.  Well-child visit rates in 2022 actually start and end lower than baseline (2019). 2023 starts off lower than 2022 ended and ends in the same place that baseline ended.  Also, a few related issues: a) I understand why 2019 was used as baseline but perhaps it doesn't make sense.  After all, as you note, COVID had a big impact. b) Exactly how is well-child visit rate calculated?  That's not actually explained in the methods. c) Why does the rate always start off lower in April and rise towards December?  And why are January-March not shown? d) Was there variation in this outcome among the 10 MCO sites that may be useful to learn from?

Response: Additional information was added on the metric, how it's calculated, and the analysis of outcomes. 

Comment 2: 2) Tables on what was learned through PDSA cycles have the greatest potential to offer insight, but currently there is no summary statement of what they say in the results.  There is some in the discussion, but summary should be moved to results and elaborated on. 

Response: Additional information has been added to clarify and expound.

Comment 3: 3) Small thing, but would be worthwhile to explain "balancing metrics" as some readers will not be familiar with that. 

Response: Additional information has been added to clarify and expound.

Reviewer 3 Report

Comments and Suggestions for Authors

Reading the abstract, I have no information about what the purpose of this article is

In the introduction, it would have been good to refer globally to the Medicaid program, which operates in the US, and show its broader background. I don't see the purpose of the survey.

The methodology lacks a detailed description of the survey's audience, as well as a description of the survey itself - the types of questions, the cafeteria of responses. In the methodology I do not see clearly described information on what the survey was about. You should rebuild this section and describe in detail the methodology, the survey recipients, the questionnaire used, etc. 

The results are presented correctly, the tables are clear - I suggest moving the table to the results section.

The discussion is prepared correctly.

Conclusions are too sparse, too general. I suggest referring more deeply to the survey results and comparing them with the literature items discussed to draw conclusions. Practical implications are also missing.

Author Response

Comment 1: Reading the abstract, I have no information about what the purpose of this article is

Response: Additional information added to provide clarity.

Comment 2: In the introduction, it would have been good to refer globally to the Medicaid program, which operates in the US, and show its broader background. I don't see the purpose of the survey.

Response: Line 34 states Medicaid and the Children's Health Insurance Program (CHIP) are the public insurance programs offered in the United States of America.

Comment 3: The methodology lacks a detailed description of the survey's audience, as well as a description of the survey itself - the types of questions, the cafeteria of responses. In the methodology I do not see clearly described information on what the survey was about. You should rebuild this section and describe in detail the methodology, the survey recipients, the questionnaire used, etc. 

Response: Additional information added to provide clarity.

Comment 3: Conclusions are too sparse, too general. I suggest referring more deeply to the survey results and comparing them with the literature items discussed to draw conclusions. Practical implications are also missing.

Response: Additional information added to provide clarity.

Round 2

Reviewer 1 Report

Comments and Suggestions for Authors

The authors have not addressed the comments made previously.

Author Response

Comment: The authors have not addressed the comments made previously.

Response: See edits in attached document.

Reviewer 2 Report

Comments and Suggestions for Authors

Explanation of balancing metrics has been provided. Thanks.

An explanation of the outcome metric has now been provided.   However, if I am understanding it correctly, this explanation raises a new question.  It seems from the explanation that the metric is cumulative over the course of the year. So, the final statistic is really the December figure. That is exactly the same for the baseline and intervention year, so it would seem that there was no intervention impact.  (On the other hand, maybe I am misunderstanding what cumulative means, since, in a couple of spots, the rate goes down from one month to the next, suggesting that it is not cumulative in the sense that I understand.)  What does seem to be the case is that the rate mounted more quickly in the intervention year than in the baseline year, perhaps indicating that more people got the vaccines on time, even though the final rate was the same?  Clearly need further clarification both of what this metric is measuring and what the results mean. 

I do not see any text added summarizing the survey tables in the results.  

Comments on the Quality of English Language

fine

Author Response

Comment 1: An explanation of the outcome metric has now been provided.   However, if I am understanding it correctly, this explanation raises a new question.  It seems from the explanation that the metric is cumulative over the course of the year. So, the final statistic is really the December figure. That is exactly the same for the baseline and intervention year, so it would seem that there was no intervention impact.  (On the other hand, maybe I am misunderstanding what cumulative means, since, in a couple of spots, the rate goes down from one month to the next, suggesting that it is not cumulative in the sense that I understand.)  What does seem to be the case is that the rate mounted more quickly in the intervention year than in the baseline year, perhaps indicating that more people got the vaccines on time, even though the final rate was the same?  Clearly need further clarification both of what this metric is measuring and what the results mean. 

Response 1: See edited version with additional information.

Comment 2: I do not see any text added summarizing the survey tables in the results.  

Response 2: See edited version with additional information.

Reviewer 3 Report

Comments and Suggestions for Authors

still have not been corrected comment 2 and 3 on:  

Comment 2: In the introduction, it would have been good to refer globally to the Medicaid program, which operates in the US, and show its broader background. I don't see the purpose of the survey.

Response: Line 34 states Medicaid and the Children's Health Insurance Program (CHIP) are the public insurance programs offered in the United States of America.

- This 1 sentence is not enough, I was clearly asking for a reference to health insurance programs and background in the US. 

Comment 3: The methodology lacks a detailed description of the survey's audience, as well as a description of the survey itself - the types of questions, the cafeteria of responses. In the methodology I do not see clearly described information on what the survey was about. You should rebuild this section and describe in detail the methodology, the survey recipients, the questionnaire used, etc.

Response: Additional information added to provide clarity. 

- Where? I don't see any correction to the methodology in the text except for the addition of the word 'conducted'.

This is incomprehensible to me.

Author Response

Comment 1: This 1 sentence is not enough, I was clearly asking for a reference to health insurance programs and background in the US. 

Response 1: See edited version with additional information.

Comment 2: Where? I don't see any correction to the methodology in the text except for the addition of the word 'conducted'. This is incomprehensible to me.

Response 2: See edited version with additional information.
